# TP53 and the Ultimate Biological Optimization Steps of Curative Radiation Oncology

**DOI:** 10.3390/cancers15174286

**Published:** 2023-08-27

**Authors:** Anders Brahme

**Affiliations:** Department of Oncology-Pathology, Karolinska Institutet,17176 Stockholm, Sweden; andersbrah@gmail.com

**Keywords:** cell survival, low-dose hypersensitivity, dual double-strand breaks, TP53, low-dose apoptosis, APR246, radiation time–dose fractionation, radiation dose–response relationships, light ion radiation therapy, radiation therapy optimization

## Abstract

**Simple Summary:**

This review of several recent developments and findings in molecular and radiation biology and medical physics was put together to improve the treatment of cancer. An improved understanding of the effect of radiation on tumors and normal tissue responses makes it possible to treat tumors more effectively and with less damage to normal tissues. This paper is based on a new, improved understanding of the cellular damage and repair processes and the associated internal cellular program to eliminate itself when critically damaged (apoptosis) to avoid severe mutations of the hereditary material of the individual. An improved radiation dose–time fractionation method is proposed to more effectively cure cancer without damaging the surrounding normal tissues. Furthermore, the best possible radiation beams of intensity-modulated electrons, photons, and the lightest ions from lithium to boron are proposed to obtain the best possible treatment results, increasing the tumor cure probability by about 20%.

**Abstract:**

The new biological interaction cross-section-based repairable–homologically repairable (RHR) damage formulation for radiation-induced cellular inactivation, repair, misrepair, and apoptosis was applied to optimize radiation therapy. This new formulation implies renewed thinking about biologically optimized radiation therapy, suggesting that most TP53 intact normal tissues are low-dose hypersensitive (LDHS) and low-dose apoptotic (LDA). This generates a fractionation window in LDHS normal tissues, indicating that the maximum dose to organs at risk should be ≤2.3 Gy/Fr, preferably of low LET. This calls for biologically optimized treatments using a few high tumor dose-intensity-modulated light ion beams, thereby avoiding secondary cancer risks and generating a real tumor cure without a caspase-3-induced accelerated tumor cell repopulation. Light ions with the lowest possible LET in normal tissues and high LET only in the tumor imply the use of the lightest ions, from lithium to boron. The high microscopic heterogeneity in the tumor will cause local microscopic cold spots; thus, in the last week of curative ion therapy, when there are few remaining viable tumor clonogens randomly spread in the target volume, the patient should preferably receive the last 10 GyE via low LET, ensuring perfect tumor coverage, a high cure probability, and a reduced risk for adverse normal tissue reactions. Interestingly, such an approach would also ensure a steeper rise in tumor cure probability and a higher complication-free cure, as the few remaining clonogens are often fairly well oxygenated, eliminating a shallower tumor response due to inherent ion beam heterogeneity. With the improved fractionation proposal, these approaches may improve the complication-free cure probability by about 10–25% or even more.

## 1. Introduction

Two recent DNA-repair-based publications on the accurate quantification of cellular survival and damage to tumors and normal tissues [1,2] have significantly improved our ability to precisely describe responses to low and high radiation doses and ionization densities, or more precisely, linear energy transfers (LETs). The accuracy is far beyond the possibilities of the traditional linear quadratic (LQ) survival model and allows for the estimation of the apoptotic fraction, as indicated in Figure 1. Not only are the undamaged cells separated from the sublethal damaged cells but the two major DNA damage repair pathways, namely NHEJ (nonhomologous end-joining) and HR (homologous recombination), are also separately identified, which is particularly valuable in beams with high-LET components. Furthermore, the major forms of interaction between them, such as independent homologous and nonhomologous repair, as well as the homologous repair of nonhomologous misrepair, are accounted for, and so is the probability of inducing apoptosis, and potentially senescence and other cell cycle losses [1,2]. As shown in the lower part of Figure 1, the key cross-sections of the RHR formulation are *σ*_n_, *σ*_h_, and *σ*_i_. These are the key events induced by radiation for the induction of nonhomologous end-joining and homologous recombination repair pathways and direct inactivation, and for simplicity, in Figure 2 (cf also Section 2.4 Figure), they are denoted as *n*, *h*, and *i,* respectively. Interestingly, this ability to quantify apoptosis has helped us to identify the early low-dose hypersensitivity (LDHS) and low-dose apoptosis (LDA) of most normal tissues but also tumor tissues with intact TP53 and ATM genes [2]. This important mechanism, probably developed by the survival advantage of the species, ensures minimal risk for severe mutations before the DNA repair system is fully functional after about ½ to 1 Gy (cf. Figure 1 and Figure 2 and [1,2,3,4,5]). As a compensating measure, the apoptosis-inducing caspase3 remarkably “remembers” this low-dose apoptotic cell loss and starts cellular repopulation to re-establish homeostasis in the tissues after being irradiated. This useful mechanism in normal tissue is a well-known problem after suboptimal radiation therapy as it can cause accelerated tumor cell repopulation at the end of a noncurative treatment [6]. A clear curative intent is probably the principal way to avoid this tumor-reactivating mechanism. The above studies also identified that the most effective HDA is induced through Ser 46 phosphorylation, i.e., via ATM and p38K at dual double-strand breaks (DDSBs) generated by the lowest LET ions, largely as they have the highest fluence of secondary *δ*-electrons generated by the primary ion fluence per unit dose [1,2]. With a very high LET (of carbon ions and others), the apoptosis and senescence will instead also be high in the normal tissues in the entrance region of the beam, which is undesirable from a complication-free cure point of view, even if hypoxic tumor-cell inactivation may be marginally improved by such a process [1] (Figure 22), [7,8,9].

## 2. TP53 and Cell Survival and Apoptosis at Low and High Doses and LETs

### 2.1. TP53 Damage Response

The new increased flexibility of describing the shape of the cell survival curve has resulted in a significantly improved description of low- and high-dose apoptosis and LET-radiation cell survival in mutant, wild-type, and repair-gene knockout cells. It is well known that the nonhomologous end-joining (NHEJ, [1,2]) pathway is the dominating DNA repair process of low LET, and it is very fast. Ku70, Ku80, and a DNApk dimer bind together the broken DNA ends in a few seconds and simultaneously recruit p53 [10], such that high-dose and LET local damage can also be correctly repaired. This process is essential at high LET levels when the MRN dimer complex often replaces the Kus and DNApk if the cell is in the S or G2 phase of the cell cycle, and it may need the homology-searching mechanism of homologous recombination (HR) for high-fidelity repair [14,16]. This makes HR more important than NHEJ at very high LETs, partly as less low-LET-type damage is induced ([2] Figure 8 with three ions (B, C, and N) and three cell lines, [17]: C, [18]: C), even if it is not very clearly and slightly indirectly observed, as in the case of the last two references. Most interestingly, the new repair formulation allows for the approximate quantification of concurrent independent NHEJ and HR repair, as well as the HR repair of NHEJ misrepair, alone or concurrent with other ongoing HR repair processes. These newly established DNA repair terms (cf. lower right corner of Figure 2) make it possible to describe cellular repair far beyond the conventional linear quadratic model (LQ). Interestingly, the new DNA-repair-based formulation inherently describes LDHS and LDA as they are linked to the DNA repair system of most, if not all, normal tissues, as described in more detail in Figure 1. This figure illustrates how the TP53 gene works as a complex cellular controller by determining how the DNA damage structure should best be repaired, and whether senescence and apoptosis are needed, as shown in Figure 1 and Figure 2 [1,4,10,11,12,19]. After DNA damage by radiation, it works principally through the augmented phosphorylation of three serine sites (16, 20, 46 [11]) following the increase in the extent of damage through some of its key upstream proteins such as ATM, CHK2, and p38K as well as 53BP1 or BRCA1,2 to signal whether the currently most optimal repair process is NHEJ or HR. Most tumor cell lines suffer from a TP53 mutant gene, making the early/low dose (0.5–1.5 Gy) radiation response more gradual without LDHS and low-dose LDA, as shown in the cell survival insert in Figure 1 (cf also Section 2.4 and Section 2.6 Figures below). As TP53 intact cells are irradiated after about ¼ Gy, ATM is auto-phosphorylated and in turn phosphorylates the serine 15 site on p53 to start NHEJ repair. After a total of ½ Gy or about 18 DSBs, CHK2 is also phosphorylated and phosphorylates the serine 20 site on p53 to achieve fully efficient DNA repair with both HR and NHEJ [1,2,3] around 1 Gy. As seen from the lower left insert in Figure 1, this last step results in a switch in normal tissue sensitivity from an initial LDHS stage, before the full serine 15 and 20 phosphorylation of p53 generates a more radiation-tolerant state. After that, the cellular repair system is fully activated and functional, with reduced cell loss and almost a survival plateau around 2 Gy [1,2,4,20,21,22]. It is clear from Figure 1 and Figure 2 and the Graphical Abstract that tumor doses well above 2 GyE are needed for cure, and the dose should be just above 2 Gy or below to ensure optimal normal tissue radiation recovery, as clearly shown in Figure 2 with the pale dotted tangent line through the origin. The LDA and LDHS of normal tissues are caused by about 5–15% acute low-dose apoptosis (Figure 1 and Figure 2 and [4,5]), but interestingly, most likely, due to the compensating measure of caspase-3-induced cellular repopulation [6], late effects are few. This will re-establish homeostasis in normal tissues and thus minimize late normal tissue damage, but it may sometimes also repopulate malignant tumor clonogens if they are not entirely eradicated by the treatment [6]. This means that LDA really protects normal tissues from potential low-dose mutations before NHEJ and HR are fully functional and can address the damage (cf. Section 2.2). Furthermore, in terms of radiation therapy, it means that the fully functional DNA repair system should continue to be utilized until the more severe high-dose apoptosis sets in after about 2–2.5 GyE, as seen in Figure 1. Already, we understand that there is an optimal radiation therapy fractionation window in normal tissues around 1.8–2.3 Gy/ Fr to minimize normal tissue damage, as discussed in further detail below (cf. Section 2.5).

### 2.2. Cell Survival

Interestingly, a recent publication on DNA repair [2] (Figures 7, 9a,b and 12a,c,d) demonstrated that this early low-dose cell loss before establishing full repair efficiency is mainly due to p53 LDA induction, in general agreement with [4,5] and the present study’s Figure 1 for a TP53 intact tumor. This direct LDA and LDHS is most likely a general property of intact normal tissues and the cell’s natural way to protect itself from low-dose, potentially cancerous, mutations before full repair efficiency is established [2,4,22]. Standard therapeutic 2 Gy fractions generate 75 DSBs and possibly one or two dual DSBs (DDSBs) that may be inducing high-dose apoptosis (HDA) via serine 46. As seen from the insert in Figure 1, many tumor cell lines, and more than 50% of all tumors generally, have a mutant TP53 gene that commonly eliminates most of LDHS and early LDA to obtain a radiation-resistant LQ-like shoulder. As shown for mouse embryo fibroblasts with key repair genes knocked out, both CHK2-/- and particularly ATM-/- cells lose all the LDHS of wild-type (wt) cells, highlighting the importance of the low-dose phosphorylation steps [2] (Figure 13) and [23]. Obviously, there are also some wt TP53 and wt ATM tumor cell lines that may show the LDHS property (cf Section 2.6 Figure below). The most recent RHR paper also investigated LDHS and LDA for light ions, demonstrating that it generally peaks at low-LET ions (≈20 eV/nm, see Figure 3), where, for reasons related to ion interaction physics, the highest low-energy δ-electron production per unit dose occurs, as the LET is low but sufficient to induce apoptosis, and the ion fluence per unit dose is highest. Thus, apoptosis peaked here (31% measured at 3 Gy of 40 eV/nm B^5+^ [2] (Figures 7, 9, 10 and 12)), and a clear but small LDHS was observed [24] mainly due to the early dual NHEJ only and HR misrepair (5% at 0.5 Gy, calculated [2] (Figures 7, 8 and 9b)). The 31% value was not just LDA but mainly serine-46-induced HDA. So, even if there is a weak low-LET ion LDHS and LDA, it is most likely not sufficient to establish a real light ion fractionation window with carbon ions but surely with the entrance plateau and fragmentation tail of low-LET lithium ions (in Figure 1 [2,7], in Figure 2 [2,20] (with ^60^Co) and Figure 4 (with mut p53-reactivated)), as well as for lowest-LET boron ions and ^60^Co in Section 2.6. The Bragg peak should always be reserved for target tissues. Thus, HDA is very valuable [25], and even more so, the often associated senescence [1,2,26,27,28], also described in Figure 1, since it does not cause later problems with caspase-3 [6]. To further elucidate the development of the shapes of the cell survival curve, some key steps are summarized in Figure 2, considering the normal lung epithelial cell data [1,2,20]. During the 1940s to 1960s, the logarithmic linear model (Ln) was established, with a back extrapolated (slope *D*_0_) initial cell number (*n*) larger than 1 to account for cellular division and repair during irradiation. Alternatively, the so-called quasi-threshold dose (*D*q) was also used to indicate that the linear extrapolation seemed to start from a dose that was wasted due to tumor cell repopulation and repair. Clearly, this was a rather crude way to describe the early cell survival that often was of low clinical importance at the time but not really today [4]. From the late 1960s, the still currently dominating linear quadratic model could better describe the slight curvature of the quasi-linear high-dose cell survival by its α and β factors of most tumors but did not account so well for the low- and high-dose survival, at least for most normal tissues, as seen in Figure 2. It also misses a true repair term, as a high repair requires a high β, but that means less survival since the term is negative, so α needs to be reduced to better describe the survival, which may only work in a small-dose region and α loses its original meaning. This illogical effect, and the fact that it gives a poor description of LDHS normal tissue survival, has misled two generations of radiation biologists to trust it rather uncritically. The beauty of the third repairable–conditionally repairable (RCR) model is that it is very simple and describes what occurs at large to the cells using Poisson statistics with a simple exponential term for missed cells and a linear exponential term for the correctly repaired fraction of sublethal hit cells. Thus, it is a logical continuation of the Ln expression, as seen in Figure 2 and [29]. It therefore can describe the LDHS quite well and solves the repair and overkill problem at high doses seen with the LQ model. The repairable–homologically repairable formulation (RHR) goes a few steps further by accounting for the two major DNA repair pathways, as mentioned above, and their associated misrepair processes, as seen in the lower right corner of Figure 2. This model can therefore handle, e.g., cell lines with mutant and/or knocked-out repair genes, high and low LETs, and apoptosis induction [1,2]. It is thus extremely important to consider the significant differences between the cell survival of most tumors and generally LDHS normal tissues when designing optimal radiation therapy protocols. It is unfortunate that the bulk of established tumor cell lines mostly suffer from TP53 and associated mutations, so they can easily grow in the lab and lead to the assumption that all cell lines have LQ-like shoulders, almost making the LQ model a dogmatic model of true cell survival. In fact, it is most likely that all intact normal tissues have wild-type TP53 and ATM genes, and thus are linked to LDHS and LDA, as recently indicated [1,2,4], and it is probably an inherited growth advantage to avoid cancerous transformations after low-level genetic damage that may not be correctly repaired. The intriguing reason why it went undetected for such a long time is that too few studies were conducted on live normal tissues at low doses and with sufficient accuracy (until Joiner [30]), but also because the associated accelerated repopulation via caspase-3 tries to compensate for LDA and LDHS at the end of irradiation to re-establish tissue homeostasis [6]. A relative apoptotic effectiveness (RAE) value of about 3.4 has been noted for low-LET boron ions around 40 eV/nm, whereas the peak relative biological effectiveness (RBE) was found to be about 3.5 but closer to an LET of about 160 eV/nm, as seen in Figure 3 (cf also Section 2.6 Figure).

### 2.3. Apoptosis Induction

The new increased flexibility to describe the shape of the cell survival curve and the major forms of interaction between HR and NHEJ, such as homologous repair of nonhomologous misrepair, are accounted for, and so is the probability of inducing apoptosis as a result of nine different types of misrepair processes [2]. The increase in apoptosis with doses at varying LETs provides a very good fit to the experimental data ([2] (Figures 7, 9, 10 and 12), mean error <≈0.5%). Interestingly, dual concurrent HR and NHEJ misrepair mechanisms in the same cell nuclei dominate at medium LETs, whereas concurrent and only HR misrepair mechanisms fully dominate at severely high LETs, as practically all types of such damage require HR processing [2,7,14]. In low-LET ^60^Co, almost all misrepair processes contribute via NHEJ that now dominate cell repair (see [2] for more details). It is indeed seen that low-LET ions produce the highest LDA and HDA per unit dose since the fluence and the number of ions are reduced by half each time the LET is doubled, and so is the apoptosis, as described in more detail in [2] (Figures 7, 9, 10 and 12). The upper right corner of Figure 3 includes the theoretical expression for apoptosis induction showing the increase by lowering the LET by its last 1/LET term that breaks down when the LET is too low, so no apoptosis can be induced. From Figure 3, it is seen that lower LETs (≈20 eV/nm) are preferable to optimize tumor apoptosis and senescence simultaneously, as they are also lowered in normal tissues, and this may generally be better than optimizing the total relative biological effectiveness (RBE) in the tumor and consequently maximizing the hypoxic cell kill. However, unfortunately, this is associated with adverse effects on normal tissue damage in front of and behind the tumor [1,2,9,11], which poses clinical challenges when using Neon ions [31] but also to some extent Carbon ions. The optimal therapeutic choice may also depend on the use of adjuvant therapies, e.g., to enhance antitumor immune reactions or ROS effects [2].

### 2.4. Reactivation of Mutant TP53

The cell survival effect of the mutant p53-reactivating compound PRIMA-1 on the TP53 mutant SCLC cell line U1690 after exposure to ^60^Co γ-rays is shown in detail in Figure 4, indicating full survival. The curves with PRIMA-1 are here shown as solid lines, and the plain SCLC cell line U1690 without PRIMA-1 are shown as dashed lines, with arrows indicating the change in survival by the presence of PRIMA-1 during irradiation. At higher doses, the relative survival loss is quite large (almost 27% at 4 Gy, see Figure 4). Therefore, there would be a clinical advantage of using high doses per fraction with PRIMA-1. There is an increased HR repair alone (*f*_h_ by almost 50%) but also to fix NHEJ misrepair (*l* is more than doubled), indicating that the HR pathway is largely improved through p53 reactivation, partly increasing the very low-dose apoptosis [2]. The recovery of the HR pathway with PRIMA-1 is more than tenfold increased below 0.15 Gy, sixfold at about ½ Gy (LDA), fourfold at 1 Gy, and more than doubled at high doses (HDA) with PRIMA-1. Due to the wide therapeutic effect spectrum of the active component MQ of APR 246 and PRIMA-1 [2,32], both on reactive oxygen species (ROS), inhibiting the enzyme thioredoxin reductase 1 and thioredoxin and decreasing cellular glutathione levels and increasing apoptosis, it is difficult to say exactly what caused the increase in HDA apoptosis by ≈15% in the current study. Even if PRIMA-1 increased the HR only, and its repair of NHEJ misrepair, this probably occurred due to improved HR initiation via TP53, there are also indications that apoptosis could be caused by direct mitochondrial effects or via caspase-3, and the cell cycle block via p21 is not generally restored either [33,34]. This latter fact may even be an advantage for cancer treatment as tumor cells will continue cycling and incorporate damaged DNA in their genomes without repair, and they may finally end up in a mitotic catastrophe situation like in classical radiation therapy. The first very interesting cell culture study of using APR 246 on colorectal cancer combined with radiation was recently published, and mut TP53 showed almost wt TP53 response with 5 μM APR 246, whereas TP53 Null cells were only half as responsive [35] generally consistent with Figure 4. At 7.5 μM, the mut cell line was even more responsive than the wt, which should be a valuable treatment property. In a tumor growth assay, 20 μM APR 246 and 6 Gy both halved the mut xenograft size, whereas the combination brought it down to ≈1/5, but wt showed only a 30% APR 246 reduction, and null and mut cells were similar [35]. Interesting Venn diagrams of significantly enriched pathways and genes for combined and radiation-alone treatments were also included for wt, mut, and TP53 null cells.

### 2.5. The Fractionation Window

Interestingly, during the 125 years of curative radiation therapy, we have already found how to fractionate radiation treatments to maximize curability using the well-established 2 Gy/Fr dose regiment. This largely happened by trial and error, even if the mechanism may not have been fully understood until recently. According to Figure 1, this is, to a significant part, due to the LDA of most normal tissues, and it is essential to maximize the use of it in cancer treatment to maximize the normal tissue tolerance. Normally, the 2 Gy dose is prescribed to the tumor, and more specifically the internal target volume [36], as the fractionation widow was established in the era of parallel-opposed beams. However, the recognition that most normal tissues are low-dose hypersensitive, and most tumors are not, as well as the fractionation window between about 1.8 and 2.3 Gy, shown in Figure 1, Figure 2 and Figure 5 (for details, see [1,2,37,38]), necessitates the reconsideration of the 2 Gy/Fr prescription, which is more precisely described in the clinical conclusions in Section 4. Furthermore, it is important to point out that the existence of a ≈ 2 Gy/Fr established optimal treatment regiment, which was identified some 80 years ago and may be expressed differently as the existence of a “fractionation window” where radiation therapy works well. This is in fact the most significant clinical proof for the general existence of low-dose hypersensitivity in most normal tissues not least because it was established in the era of parallel-opposed beams, with almost equal doses to the tumor and organs at risk. In addition, the molecular mechanisms seen in Figure 1 explain how it works in finer molecular details, and the new radiation biology in Figure 2 shows how the ≈2 Gy/Fr tangent to the survival curve from the origin (pale dotted line in Figure 2) reflects the shallowest and least damaging irradiation effect on normal tissues when having to deliver high doses to deep-seated tumors. In Figure 5, the effect of varying the dose per fraction on normal tissue damage is illustrated based on the experimental survival data in Figure 2 and more clearly indicates the function of the clinical fractionation window in reducing normal tissue injury. Both lower (yellow) and higher (violet) doses per fraction cause a higher probability of damage, so it is preferable to stay within 1.8–2.3 Gy/Fr, at least when passing through the lungs. Unfortunately, the conventional LQ model is blind to and does not describe this low-dose-per-fraction damage in normal tissues (except for Joiner’s nice but complex modification, handled better using the much simpler RCR formula, Figure 2) and of course also with the more complex and exact RHR formulation [1,2], which explains what really occurs and also eliminates other LQ shortcomings.

The fractionation window is important to consider in radiation therapy optimization in order to eradicate a tumor with minimal normal tissue damage: a too-low dose/Fr (<1.5 Gy) causes more accumulated LDHS and LDA in normal tissues, and a significantly higher dose than ≈ 2.5 Gy may result in more “double trouble” with both a high dose and high dose/Fr, considering the fact that the same curative dose need to be imparted to the gross tumor in both cases.

### 2.6. Secondary Cancer Induction

To further illustrate the power of quantifying apoptosis, in Figure 6 the probability of inducing a secondary cancer is illustrated based on experimental cell survival and apoptosis data [2] (Figures 7 and 9). It is unlikely that the apoptotic fraction will contribute to secondary cancer induction (except possibly in TP53 mutant cell lines), so it is useful that this fraction can be estimated using the new RHR formula and removed from other forms of misrepair to more accurately describe the cells that are potentially capable of generating secondary cancer. This cell fraction, as seen in Figure 6, has its peak in the 1–4 Gy range, so in radiation therapy optimization, it is desirable to minimize this volume as much as possible in normal tissues. Figure 6 also shows that the maximal risk is the smallest for low-LET ions (blue-shaded) largely due to their high apoptotic induction. The real secondary cancer risk may be in the order of 5% of the maximal values in Figure 6 or less. Obviously, these experimental data are not really relevant for the surrounding normal tissues that may receive a fair amount of dose and are at risk for secondary cancer. However, the present tumor cell line is at least wt TP53 so probably not the most extremely mutated one and may, in first approximation, be assumed to represent both normal and tumor tissue risks. Furthermore, the dose axis is clearly the dose per fraction, so it means that the total dose is increased by the number of fractions. Interestingly, the curves can also be used to optimize the dose per fraction, and then a higher dose per fraction of higher LETs (e.g., 2 Gy at 160 eV/nm corresponding to ≈6 GyE) may even be better than the lowest peak value at 40 eV/nm. In fact, if the plot is instead drawn as a function of dose equivalent (dose x RBE; the 50% survival RBEs are given in the figure), all the maximal doses align very well with their present maximum values unchanged, showing that the ion with the lowest LET will always minimize the secondary cancer risk for a given delivered dose equivalent. Notably, this secondary cancer risk is a contraindication for large, low-dose volumes with many beam portals in intensity-modulated radiation therapy using methods such as “rapid arc”, “volumetric arc”, and “tomotherapy” on non-seniors, which may have time to develop secondary cancer after some 20 years [39,40].

### 2.7. Simplistic Clinical Example

Let us assume a crossfire scenario in which two perpendicular high-energy photon beams intersect to deliver a classical 60 Gy to the tumor in 30 fractions. This implies that the normal tissues receive about 1Gy from each beam alone. To comply with the present approach, we should instead deliver about 2 Gy per beam to receive 4 Gy in the tumor, and we will need only 15 fractions to receive 60 Gy. However, 4Gy/Fr in the tumor is likely to result in much more tumor kill than 2 Gy/Fr, so we may only need ≈10–12 fractions and 40–48 Gy. Alternatively, we could reduce the dose per fraction to 3 Gy and use 20 fractions for 60 Gy total. Probably, 18 fractions or a little less may suffice with 54 Gy total. Obviously, the reduction in the total dose could be estimated using the classical LQ method, assuming a TP53 mutant tumor, whereas the normal tissue damage will require an RCR or RHR calculation. Classically, we obtain a 0.7^30^ ≈ 0.000023 accumulated normal tissue survival reduction (Figure 2) whereas RHR would lead to a 0.5^15^ ≈ 0.000031 or preferably 0.65^18^ ≈ 0.00046, i.e., ≈20-fold better normal tissue survival, thus very likely increasing the complication-free cure. Interestingly, a somewhat similar thinking was recently presented by Yarnold and coworkers [41]. There are therefore definitely many reasons to re-evaluate the time–dose fractionation along the current ideas and those presented in a previous study on DNA repair ([1] (Figure 21); see also the Graphical Abstract) to really introduce a major paradigm shift in curative radiation therapy thinking as suggested in the Section 5.

## 3. Influence of Microdosimetric Beam Characteristics on the Dose–Response Relation of Tumors and Normal Tissues

### 3.1. The Dose–Response Relation

The shape of the tumor dose–response relation is accurately described using the binomial or Poisson statistical probability for having no viable surviving tumor clonogens at the end of the treatment, which is expressed as follows: *P*_B_(*D*) = e^−*N*_0_ · ^*^S^*^(*D*)^ = e^−*N*_0_ · e^−*D*/*D*_0_^^(1)
where the last step is the simplification possible with a constant dose per fraction *D*, *N*_0_ is the initial clonogen number, *S*(*D*) is the relative clonogen survival after dose *D*, and *D*_0_ is the exponential slope defined in Figure 2. As dose *D* is increased, the number of remaining clonogens is reduced until at high doses, the number of surviving clonogens tends to zero, and the cure probability approaches unity along a sigmoidal curve, as shown in Equation (1) and Figure 7. The curve shape is reminiscent of the cumulative distribution function of a random variable, which, by definition, also starts from zero, finally reaching one or 100% when all random events have been counted. Interestingly, the curve shape is rather well described (within a few %) using the cumulative generalized gamma distribution, but even more exactly using the perhaps more well-known extreme value distribution, which is known to describe the distribution of outliers of random processes, whereas the rest of the distribution is approximately normal distributed.

In fact, radiation therapy is probably a perfect example of the extreme value distribution, as it is well known that the last few most radiation-resistant tumor clonogens survive the initial major part of the treatment (≈60Gy/70Gy ≈ 85%) without being killed and remain to form the tumor control curve, as recently described in more detail [1] (Figure 20). It is therefore not surprising that Equation (1) can be rewritten to perfectly fit the cumulative extreme value distribution as follows:e^−e(*μ*−*D*)/*ν*^ = e^−e(*D*_0_∗^^ln*N*_0_−*D*)/*D*_0_,^(2)
where the last part is a rewriting of Equation (1), and therefore the approximate mean value of the extreme value distribution *μ* = *D*_0_ln*N*_0_ and the “radiation resistance” *ν* = *D*_0_ can be identified in Equation (2). More precisely, the true mean value is actually $\bar{D}$ = *μ* + *ν* ∗ *γ* = *D*_0_(ln*N*_0_ + *γ*), the median value is *D*_50_ = *μ* − *ν* ln(ln2) = *D*_0_ln(*N*_0_/ln2), the Variance is *V* = *σ**_D_*^2^ = *π*^2^*D*_0_^2^/6, and finally, the relative standard deviation is *σ_D_/*
$\bar{D}\;$= *π /*$(\sqrt{6}$ (*μ/ν* + *γ*)) = *π /*$(\sqrt{6}$(ln*N*_0_ + *γ*)). These are important parameters from a microdosimetric point of view (in all these equations, *γ* ≈ Euler’s gamma constant, not *γ*_C_). For a common tumor size of *N*_0_ = 10^7^ clonogens, the relative standard deviation is *σ_D_ /*
$\bar{D}\;$≈ 0.0768, so only about 7.7%, resulting in a relatively steep tumor control curve that is rather sensitive to microscopic dose fluctuations. This is partly due to its high Kurtosis = 5.4 independent of *μ* and *ν* as well as *N*_0_ and *D*_0_, and so is the skewness≈1.1395, explaining the steeper rise in the tumor control curve at low doses and the shallower extended shoulder at high doses, which makes it generally quite hard to achieve 100% perfect tumor cure. 

### 3.2. The Dose–Response Steepness

A further parameter of clinical importance is therefore the normalized steepness of the clinical dose–response relationship *γ*_C_, which is defined as
(3)$$\gamma_{C}=D\frac{dP\left(D\right)}{dD},$$
and describes the fractional change in tumor cure (*γ*_C_ %) for 1% increase in the delivered dose ([42], *γ*_C_ is not *γ*!). The absolute steepness of the dose–response relationship has its maximum exactly at *D*_max_ = *D*_0_ ln *N*_0_ with *γ*_C_ = ln *N*_0_/e. Since *D* also increases at this point, the true *γ*_max_ will be at a slightly higher dose $\hat{D}$ ≈ *D*_0_(ln *N*_0_ + 1/ln *N*_0_) and thus
*γ*_max_ ≈ (ln *N*_0_ + 1/ln *N*_0_)/e,(4)
where only the first term of a power expansion is included here since it converges fast, at least for large tumors, and no exact closed analytical expression is possible. For the above example (*N*_0_ = 10^7^), the normalized steepness (*γ*_max_ ≈ 5.94) further supports the high statistical steepness observed above (see the table in Figure 7). The sharp dose–response and small relative standard deviation make the tumor cure curve quite dependent on dosimetric uncertainties. These uncertainties include both macroscopic dose variations due to imperfect beam homogeneity and the strong microscopic dose heterogeneity in light ions, which is often unavoidable, especially with the heavier ions from carbon and upwards ([43], see also [44] (Figure 3). Although it is a clear figure, it does not even show the therapeutically important ½ MGy doses in 10 nm sites of ions and their *δ*-electron cores [14] and the DDSBs at e^−^ track ends [1] due to a too-large pixel size). The recent high-resolution electron microscopic data with Ku70 binding 6 nm gold nanoparticles to detect DSBs have effectively demonstrated the common occurrence of DDSBs. For 2 MeV/u carbon ions, out of 113 DSBs, 69 were single DSBs, 22 were detected as DDSBs, 18 as triple DSBs, and 4 as quadruple DSBs [45] (Figures 2, 4 and 14). Altogether, ≈40% were of DDSB or higher complexity, most likely on the periphery of nucleosomes, where the two DNA strands are only ≈1 nm apart, with a high risk of severe misrepair at close to MGy doses in 5–10 nm *δ* e^−^ track end volumes [1] (Figures 2, 13 and 14). Interestingly, DDSBs are the most lethal cellular event for all radiations, and they are about 3 times more common in ion than in electron and photon beams [1], causing an RBE of ≈3. Of the simple ordinary DSBs, 99.2% (if not all) are effectively repaired via the NHEJ and HR pathways (Figure 1 and Figure 2 and [1,2,14]).

### 3.3. Microdosimetric Heterogeneity Effects on the Dose–Response

The microscopic hot spots naturally increase the effect on tumor cure (and normal tissue damage) through random high-dose regions, when there are still many tumor cells left, whereas cold spots delay cure at high doses, as it becomes difficult to safely hit the last few remaining most-resistant clonogenic cells. Interestingly, a switch to electrons/photons during the last week will recover a steep tumor response and a high tumor cure at a lower patient dose equivalent, as seen in the calculated Ne + e^−^ curve in Figure 7. The effect of the microdosimetric relative standard deviation *σ_D_*/$\bar{D}$ on the slope *γ*_C_ of the dose–response relation was pointed out many years ago by approximating the tumor control curve, Equation (1), using a simple error function [46,47]. In Figure 7, this is improved upon by using the full extreme value distribution according to Equation (2), now convolved with the microdosimetric relative standard deviations for e^−^ and X-rays, H^+^, He^2+^, Li^3+^, B^5+^, C^6+^, and n and Ne^10+^with *σ_D_*/=0.7%, 1.8%, 3%, 4%, 5%, 7%, and 15%, respectively for an 8 μm diameter cell nuclei (see [46] (Figure 10) and [47] (Figure 15)). The significant reduction in the *γ*_C_ value is clearly seen in Figure 8, as the microdosimetric relative standard deviation increases with increasing atomic weight, *LET*, and RBE. This problem has been well known for neutron therapy, where the relative standard deviation is almost as high as for carbon and neon ions (Figure 7) partly due to the high *LET* and low therapeutic dose (≈20 Gy). The interesting clinical neutron and photon dose–response dataset of Lionel Cohen [48] also supports this fact, as was later analyzed quantitatively and showed a reduction in the dose–response steepness by about 50% for neutrons, compared with photons [49]. This is in good agreement with the present data calculated in the tabular portion of Figure 7.

### 3.4. Treatment Optimization

The intrinsic microdosimetric heterogeneity of high-LET beams and their low therapeutic doses showed early on that a photon admixture to high-LET neutron beams was often advantageous [50,51,52,53,54]. Figure 8 further demonstrates how the microdosimetric heterogeneity not only influences the tumor cure but also naturally has a similar reduced steepness influence on the normal tissue damage curve. It is also seen that the microdosimetric variance actually increases the early rise in the tumor cure curve mainly due to microscopic hot spots in high densities of clonogenic tumor cells. However, this early advantage is rapidly lost at high doses due to microscopic cold spots when the clonogenic tumor cell densities and numbers are very low. The rather low final dose increments in a high-LET and high-microscopic heterogeneity treatment will cause microscopic cold spots where some of the few remaining tumor clonogens may survive and after treatment with caspase-3 may perform an accelerated repopulation of the tumor cells. In Figure 8, the normal tissue damage effects are shown in more detail, demonstrating that the clinical influence on normal tissues is reversed since the hot spots in more densely populated normal tissues cause increased damage and thereby further reduce the complication-free cure. Consequently, both the tumor control steepness and the therapeutic window shrink with a high *LET*, thereby reducing the probability of achieving a complication-free cure, as clearly demonstrated in Figure 8. Thus, an elevated *LET* both reduces tumor cure and increases normal tissue damage, so the selection of optimal *LET* is really critical for maximizing complication-free cure [1] (Figures 20 and 22) [7]. From this point of view, boron ions may be more advantageous than carbon at least for medium-size tumors [54]. Here, we can conclude that, at the end of the treatment, the lowest possible *LET* is optimal to minimize the dose heterogeneity, which calls for high-energy electrons or photons, as seen from the table in Figure 7. Besides the radiation modality or ion species and the tumor clonogen number, there are several other clinical factors influencing the steepness of the dose–response relation.

One of these factors is the heterogeneity of the tumor with regard to cell numbers and their sensitivities, as determined via cellular differentiation, the degree of hypoxia, and the nutritional status (Figure 7) [7,42,43,47]. Some of these heterogeneity factors, such as hypoxia, are actually reduced with an elevated *LET*, but the intrinsic ion *γ*_C_ value reduction will still limit this positive influence (see the tabulation in Figure 7; more details are provided in a recent review [54] (Figure 8.5n, 8.10, 8.23c)).

Associated with the reduced steepness of the dose–response relationship, a problem toward the end of the treatment is that the total number and density of the remaining tumor clonogens are reduced to very low values, and it is absolutely essential that even if there are few remaining clonogens, all of them need to be lethally hit to achieve total cure [1] (Figure 20). This is particularly important in the last week of a curative treatment when there is only a handful of tumor clonogens remaining, generally randomly distributed over the initial target volume. It is essential to understand that light ions above ≈lithium are no longer the optimal treatment modality since they are associated with too high of uncertainty and risk for microscopic dose nonuniformity and may not hit all the randomly located tumor clonogens with sufficient safety margin. Even if the total dose delivery was perfectly uniform until the last week, what matters is the finally remaining tumor clonogens (approximately five in a curative treatment [1] (Figure 20) and the uniformity of the final 10 GyE or five 2 Gy electron fractions, resulting in ≈86% tumor cure (→*P*_B_ ≈ e^−5*0.5^5^ see Equation (1) and Figure 2) or one 3 Gy ion fraction at RBE ≈ 3.3, providing ≈ 61% tumor cure (→*P*_B_ ≈ e^−5*0.1^ with C^6+^ survival data [1] (Figures 16 and 19)). For completeness, the Poisson probability of surviving clonogenic tumor cells after ion irradiation is quite accurately provided using the probability of no cell-nuclear (=0) hits:*P*_n_(0) *= =* e^−*σ* n^*^Φ^=* e^−*σ* n^ *^D^^ρ^*/*L*_Δ_,(5)
where $\bar{\nu }$ = *σ*_n_*Φ*, by definition, is the Poisson mean lethal hit number per cell of cross-section; *σ*_n_ ≈*πρ*^2^
*= π* 4^2^ μm^2^ ≈ 50 μm^2^ at a fluence *Φ = D/L*_Δ_ */ρ*; and *L*_Δ_ */ρ* is the restricted mass stopping power of the ions [1]. With appropriate units, this is  3.14(eVnm^−1^/Gy)∗D/*L_Δ_ /ρ*, which for 2 MeV/u carbon ions of LET ≈ 214 eVnm^−1^, and a dose of 3 Gy gives $\bar{\nu }$ ≈ 4.4 and *P*_n_(0) ≈ 1.22%, so more than 1% of the cells are not hit at this dose. Therefore, it is important to switch to a low-*LET* modality such as photons or high-energy electrons (or even protons) during the last week of treatment, with the lowest possible microdosimetric relative standard deviation in dose delivery. With a high *LET*, not only is the dose mostly concentrated on the ion tracks surrounded by microscopic cold regions (Figure 8), but also the absorbed dose is about 3 times lower, reducing the number of DSBs by a factor of ≈3 and further increasing the microscopic dose nonuniformity.

### 3.5. Optimal Use of Low-LET Beams

Interestingly, the proposed switch to a low *LET* simultaneously increases the steepness of the dose–response relation and thus enhances the therapeutic window and increases the complication-free cure through less normal tissue injury and simultaneously solving all the high-*LET* problems indicated in Figure 7 and Figure 8. This is due to the fact that there is no longer an excessive microdosimetric variance in the final dose delivery causing a reduced steepness, more normal tissue damage, and lower curative response. In retrospect, there is actually a clinical demonstration of the value of using a low-*LET* dose in addition to a high-*LET* treatment in some interesting US clinical trials with neutron therapy. Not only was it shown that the first ever gantry-mounted multileaf collimator in Seattle provided better results than ordinary block collimation, but also the use of mixed beam treatments where photons were added was found to be quite advantageous, as discussed in detail elsewhere [50,54,55,56,57,58,59]. The above discussion was not too well understood at that time, and the results could most likely have been improved even further, as discussed here, even if the somewhat better dose delivery with photons may also have contributed to the improvement. Interestingly, the optimal transition from a high to low *LET* should be when only a handful of tumor clonogens remain and most of the hypoxia is gone, preferably following the last weekend of the ion treatment, giving extra valuable HR-dependent DNA repair (see Graphical Abstract) and a tissue reoxygenation boost.

In fact, this discussion about the last week of therapy should also apply to its previous week, because the periphery of the initial internal target volume [36] may only harbor a lower density of tumor cells either due to microscopic growth or diffusion of tumor cells and/or to an added setup margin to account for organ motions and establish uncertainties [36], as seen in Figure 9 and Figure 10. Independent of the exact reason, this margin is unlikely to harbor high numbers of hypoxic or radiation-resistant tumor cells, and it therefore does not really benefit from an elevated *LET* treatment that unfortunately is likely to cause unnecessary normal tissue damage, as discussed above and seen in Figure 8. Therefore, the last part of the second-to-last week of therapy should also benefit from a switch to low-*LET* irradiation (see the few tumor clonogens in the penumbra region in Figure 9 and Figure 10), although now possibly together with a dedicated high-*LET* boost but only to the original gross tumor volume that still may harbor some hypoxic or radiation-resistant clonogenic tumor cells [46,49,50].

Interestingly, the last week and a half or so of treatment can thus be considered a low-*LET* treatment of the periphery of the four-dimensional space–time internal target volume [26], as shown in Figure 10, for simplicity and clarity drawn assuming an initial cubic tumor volume. The target volume is shown shrinking in the fourth time dimension for simplicity and clarity and to obtain a real 4D perspective. The true tumor volume may shrink a little during the treatment but not necessarily, e.g., if we really manage to induce massive senescence [7] (Figure 5) and [26,27,28]. The initial setup margin (pale pink) and the few remaining gross tumor clonogenic cells at the beginning of the last week of therapy (green volume in Figure 10) are all at the periphery of the 4D space–time internal target volume [36]. This volume will benefit from the lowest possible *LET* to secure the highest complication-free cure and steepest possible tumor response, avoiding tumor microdosimetric cold spots as well as peripheral organs at risk of hot-spot damage with high-*LET* beams (Figure 8 and [1] (Figures 20 and 22)).

## 4. Consideration of Low-Dose Hypersensitivity and Apoptosis and Photons, Electrons, and Light Ions in Radiation Therapy Optimization

The TP53 gene and its key associated DNA repair pathways, NHEJ and HR, are linked to the LDHS and low-dose apoptosis of most normal tissues, whereas most experimental tumor cell lines are rather radiation-resistant at low doses, often due to a mutant p53, as shown in the lower left insert of Figure 1 and Figure 4. This has important consequences for radiation therapy (and radiation protection for that matter), as it causes the well-known clinical fractionation window with minimal damage and apoptosis in normal tissues at ≈2 Gy/Fr of low ionization density radiations. The common tumor radiation resistance at low doses due to TP53 mutations can be treated most effectively with light ions. However, to avoid normal tissue damage, the lightest ions from helium to boron are most effective since their ionization density is mainly elevated in their Bragg peaks to be solely placed in the gross tumor, allowing for the effective use of the clinically well-established low-*LET* and LDA fractionation window at 2 Gy/Fr in normal tissues. The lower left insert of Figure 1 and (the Graphical Abstract) also implies that the low-*LET* tumor dose should be well above the 2Gy level.

Based on the above-described principles, we can now state the goals of radiation therapy more clearly and simplistically as condensed in the following key clinical conclusions:The peak absorbed dose to critical normal tissues with adverse reactions, when quasi-uniformly irradiated (organs at risk), should preferably be in the range of 1.8–2.3 Gy/Fraction and of the lowest possible *LET* and biological effectiveness (Figure 5). Interestingly, this is the dose and *LET* range that maximizes the LDHS-related normal tissue tolerance with wt TP53, as seen in Figure 1, Figure 2, Figure 4 and Figure 6 [1,2,29,46,54]. A full minimization of the total risk for complications would naturally be preferred or preferably a full so-called *P*_++_ optimization strategy approach combining 1. here with 2. and 4. below [60].In order to make the treatment as curative as possible, it is desirable that the mean dose to the tumor (internal target volume [36]) is as high as possible to ensure a true complication-free cure (*P*_+_) and perfect clonogenic tumor cell eradication. Interestingly, this can be achieved quite accurately today via advanced biologically optimized intensity-modulated radiation therapy from a few inversely planned beam directions [38,50,54,60]. This will work well even for intact TP53 and ATM pathway tumors (Figure 6 and [2] (Figure 7)) since a simple LQ-type calculation may be far from optimal.To further minimize normal tissue damage as far as possible, it is desirable to introduce an optimal weekly dose fractionation schedule where the DNA repair of normal tissues is really taken into account to minimize their injury. Up to about 50% higher tumor doses should optimally be delivered Monday morning, Wednesday midday, Friday evening, and the last evening of treatment, to use the weekend and end of therapy for maximal normal tissue recovery (see the dashed line in the Graphical Abstract, [1] (Figure 21) and [61]) and preferably still staying below the 2.3 Gy/Fr to organs at risk. This will especially optimize the weekly HR recovery towards ≈72+ h since NHEJ achieves it quite well in the 24+ h from day to day, as shown in the lower right part of the Graphical Abstract. This fractionation advantage works well for low-*LET* radiations but also for the lightest ions with mainly a low *LET* in normal tissues.For elderly patients, a larger number of optimized beam portals may be ideal, whereas younger patients may benefit from fewer beams (<5) and low-to-medium *LET* ions (see [5]) to reduce the risk for secondary cancers in extended low-dose regions (1–6 Gy total dose; see Figure 6 and [39,40]). These volumes should therefore be reduced as far as possible using sharp penumbras simultaneously as the complication-free cure (*P*^+^) or preferably the *P*^++^ optimization strategy (*P*^+^ followed by a constrained injury relaxation) are the key objectives of the treatment [60] (Figure 22).To further increase the biologically effective tumor dose delivery, a few light ion beam portals should be used preferably in the range from helium to boron ions only with their Bragg peaks located in the gross tumor volume, to keep the *LET* low (<10 eV/nm) and the dose within 1.8–2.3 Gy/fraction in organs at risk [1] (Figure 22). Organs at risk have to be passed through with beams to reach the target volume, and with the lightest ions (He-B), this can be carried out using a fairly low *LET* (<10 eV/nm). To maximize the complication-free cure, it is best to switch to electrons or photons in the last 10–15 GyE, and for bulky tumors, possibly a light ion concomitant gross tumor boost should be used in the last 5 GyE before the final plain 10 GyE low-*LET* round-up (Figure 8, Figure 9 and Figure 10; [47,54]).The influence of tumor vasculature heterogeneity on the distribution of hypoxia was carefully calculated for key tumor types and showed good agreement with clinically measured Eppendorf distributions of hypoxia [62,63,64,65]. This clinically very useful dataset for treating common hypoxic tumors with low *LET* later showed that the optimal *LET* for treating them is only as low as 25 eV/nm [46,47,54,65]. This is in good agreement with the optimal LET window of 15–55 eV/nm [1,31,54], so it also can cover other types of tumor heterogeneity and radiation resistance using helium to boron ions.For the multitude of radiation-resistant TP53 and/or ATM-mutated tumors that are often a severe clinical problem, the interesting p53 reactivating PRIMA-1 and APR-246 pharmaca may be useful to increase tumor cell apoptosis and further augment the radiation-induced reactive oxygen species effects in the high-dose tumor volume. Interestingly, PRIMA-1 and APR-246 promote the normal function of a missense mutant p53 protein-increasing LDA and HDA apoptosis in the tumor as well as senescence (Figure 4; [2,7,26,27,28,35]). Among other effects, as shown in Figure 4, it inhibits the enzyme thioredoxin reductase 1 and thioredoxin and decreases cellular glutathione levels, which is especially valuable with low-*LET* radiations, when the lightest ions are not available [2] (Figure 17).

## 5. Conclusions

The present paper introduces a new way to consider the classical 2 Gy/Fr mean tumor dose range for low-LET radiations mainly based on the recent observation that most normal tissues are low-dose hypersensitive. To minimize normal tissue damage, a dose of around 2 Gy/Fr implies optimal tolerance in normal tissues, as seen in Figure 2. This is where the least damage is generated in normal tissues, with about 1.5 Gy delivered with full NHEJ and HR repair activity, making NHEJ normal tissue recovery close to optimal, in time for the next day’s treatment fraction. Furthermore, it means avoidance of the more severe high-dose apoptosis that sets in after about 2–2.5 GyE, as seen in Figure 1. To really introduce a major paradigm shift in curative radiation therapy thinking, as described above, a dose far below or around 2 GyE/Fr should be delivered to most organs at risk to reach >≈3 Gy/Fr to the internal target volume [36]. Taking the many approaches discussed into account, the resultant increase in complication-free cure is likely to achieve improvements by as much as 10–25% and more for many tumor sites, e.g., using PRIMA-1 and APR-246 [35] for the problematic TP53-mutant tumors. About half this improvement alone was estimated to result from the improved fractionation schedule mentioned above, as described in more detail in a recent study on DNA repair (see the Graphical Abstract and [1] (Figures 10 and 21)).

## Figures and Tables

**Figure 1 cancers-15-04286-f001:**
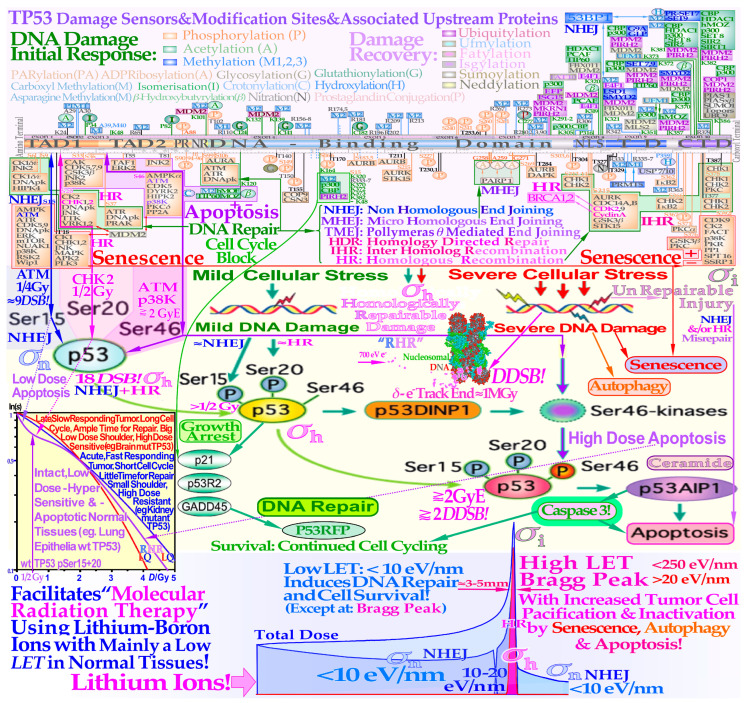
The complex response of the TP53 gene (upper third) to mild and severe genetic stress largely determines the cellular response to radiation [1,2,3,4,5,10,11,12]. Mild stress phosphorylates the serine 15 and 20 sites on p53 via ATM and CHK2, resulting in cell cycle block and DNA repair. This results in LDHS in normal tissues, but generally not in tumors, often with a mutant TP53 gene (as seen in the cell-survival insert (simplified middle third)). Local high doses or high ionization densities resulting in DDSBs (dual double-strand breaks [1,13,14]) increase the severity of the damage, also phosphorylating the serine 46 site, e.g., via p38K or ATM, and a high-dose apoptotic (HDA) response may be triggered. Lithium ions allow for unique therapeutic use by inducing a massive apoptotic–senescent tumor cell response, mainly within the Bragg peak (σ_h_ homologically repairable damage and σi direct inactivation cross-sections [1,2]). However, in front of and beyond the Bragg peak, the LET is low, and rapidly and easily repairable nonhomological damage is mainly induced (lower third, σ_n_ cross-section [1], [2] (Figure 8), [7,8]). The upper third shows the inner workings of p53 in its complex downstream pathways (cf. [15] (Figure 1)).

**Figure 2 cancers-15-04286-f002:**
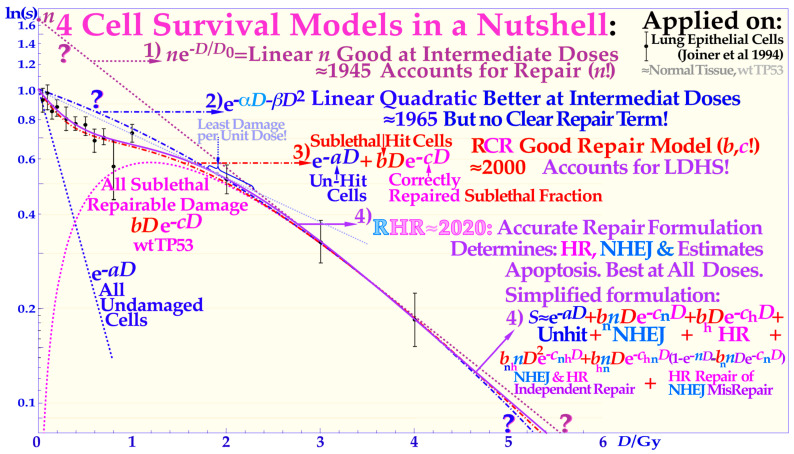
The development of models defining the shape of the cell survival curve during the last ≈ hundred years, from the linear exponential model with back extrapolated effective initial cell number (n, Ln) to the currently dominating linear quadratic formula (LQ), which does not even accurately account for cell repair as Ln does. The more recent repairable–conditionally repairable model handles the cellular repair considerably better and separates it from un-hit survival, whereas the most recent repairable–homologically repairable (RHR) formulation further accounts separately for nonhomologous and homologous recombination repair, as shown in the lower right corner, and can estimate the apoptotic fraction and individual repair processes (cf. [2] (Figures 4 and 6)). The least damage per unit dose is obtained between 1.8 and 2.3 Gy/Fr, as indicated by the fine dotted blue line with the shallowest slope possible through the unit survival point. For further details see [14] (Figure 4).

**Figure 3 cancers-15-04286-f003:**
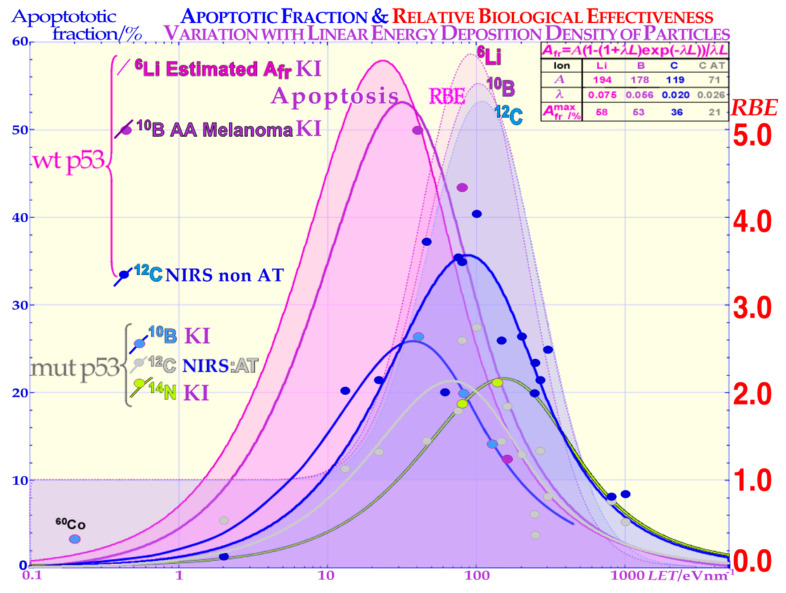
Comparison of the induced apoptosis (programmed cell death shown with solid lines and experimental dots) and the clonogenic survival measured by the RBE (pale-shaded dashed areas) as a function of increasing LET values. The induction of apoptotic cell kill depending on the status of the p53 pathway of the cells is also shown. When some part of the p53 pathway is mutant, A_Fr_ is reduced to about half its value for normal wt p53 and non-AT cell lines that are intact on the ATM gene upstream of p53. Interestingly, the A_Fr_ peaks occur at lower LET values than the RBE peaks due to the higher flux density of ions and apoptotic events per unit-absorbed dose. KI is Karolinska Institutet Stockholm Sweden, and NRIS is National Institute of Radiological Sciences, Chiba, Japan, and the table is used for extrapolation to lithium ions and discussed in the text.

**Figure 4 cancers-15-04286-f004:**
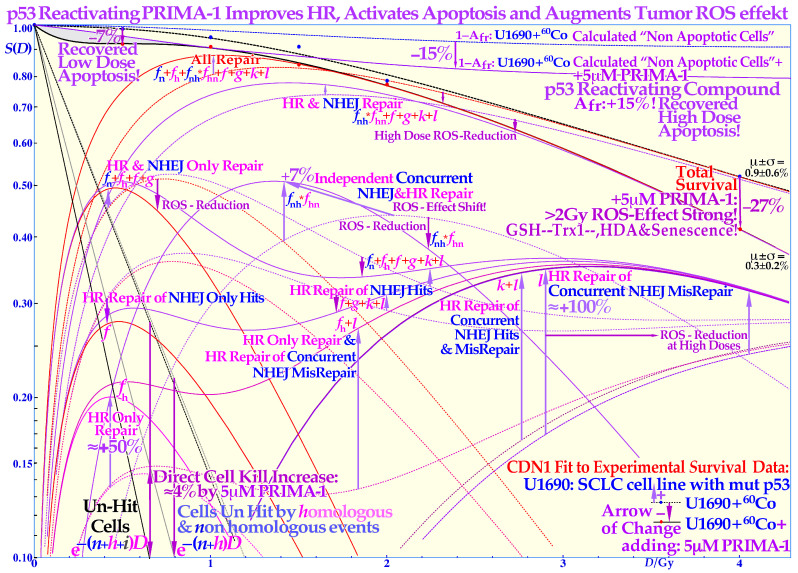
Increased LDA and HDA via mutant TP53 reactivation. The change in fractional cell survival, *S*(*D*), and major DNA repair processes with PRIMA-1 (solid lines) and without (dotted lines) for U1690 SCLC cells with mutant TP53 irradiated using 60Co γ-rays are shown. Arrows indicate the change after adding 5 μM PRIMA-1 for 14 h (10 h before to 4 h post-irradiation). Even if the apoptotic survival change is small, about 7% at low doses (LDA) and up to 15% at high doses (HDA), there are large changes in the reparability of radiation damage with PRIMA-1 added, as seen in the HR-only repair (***f*_h_** ≈ +50%) and the HR repair of NHEJ misrepair (***l*** ≈ +100%) and sum of all repair terms that even compensate somewhat for the increased apoptosis via the reactivated mutant p53. The shaded low-dose area between the calculated apoptotic cell survival and measured clonogenic survival is due to apoptotic loss before the full activation of p53 at serine 15 and 20 via the checkpoint kinases ATM and Chk2 (cf. Figure 1 and [1,2,3]). The high-dose loss in cell survival is most likely due to PRIMA-1-induced augmented toxicity through the increasing associated ROS production (PRIMA-1 inhibits the enzyme thioredoxin reductase 1 and thioredoxin and decreases cellular glutathione levels), and increasing HDA and senescence [2,21,22,23]. The mean error μ and standard deviation σ are also shown all below ≈1% but obviously much higher in some of the individual components (≈10%). The volume of data that can be estimated using the new, more flexible, and thus probably more accurate, cell survival and DNA repair formulation is striking. Updated from [2] (Figure 6) with LDA and HDA and the effects of ROS. CDN1: one-dimensional closest distance norm (not least square, see [1]). The original U1690 SCLC cell survival data were provided by Margareta Edgren at KI 2003, and the new RHR formulation significantly helped the interpretation of the wide range of effects of PRIMA-1 (*f*, *g*, *k*, *l* are repair fractions in which HR fixes various NHEJ misrepair and damage, as indirectly explained in this Figure, *f*_h_ plain HR, and *f*_n_ plain NHEJ; see [2,14] for further details).

**Figure 5 cancers-15-04286-f005:**
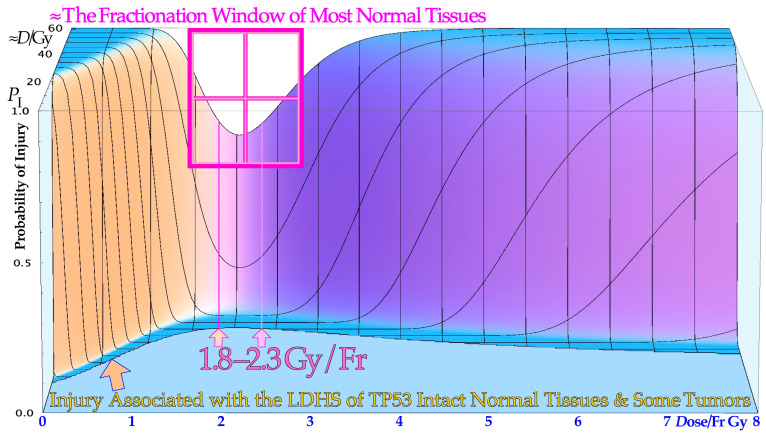
The effect of varying the doses per fraction on normal tissue damage is illustrated based on the experimental survival data in Figure 2 for lung epithelial cells. The low-dose hypersensitivity of most normal tissues establishes a therapeutic fractionation window of opportunity to cure cancer with minimal normal tissue damage. The new cell survival models discussed here can fine-tune the dose range to 1.8–2.3 Gy, as shown here for the lungs, a common organ at risk in the thorax region.

**Figure 6 cancers-15-04286-f006:**
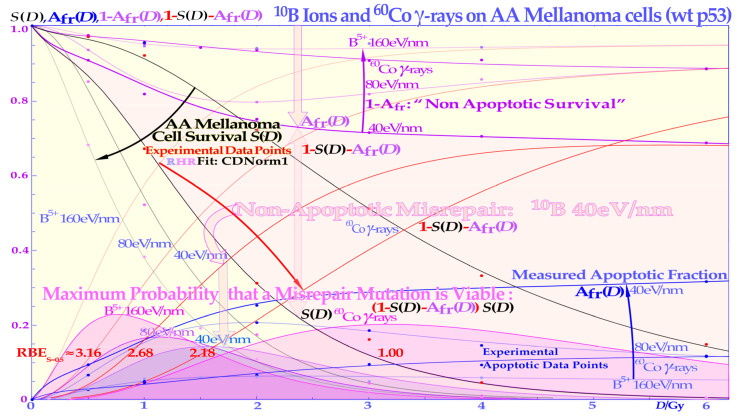
The secondary cancer induction probability as a function of the dose delivered to normal tissues. At low doses, the risk of inducing a mutation is small, whereas at high doses, the probability of generating a mutation is higher, but so is the probability of eliminating it via treatment. The risk is highest in normal tissues between 1 and 4 Gy, so this volume in patients should really be minimal. The LDA and LDHS of this TP53 intact cell line are clear from the curve shape for the two lowest LET beams. Interestingly, the risk is the smallest for the lowest-LET boron ions due to their high LDA and HDA. The upper shaded area is due to nonapoptotic misrepair for 40 eV/nm ^10^B ions. CDN1: the one-dimensional closest distance norm (not least square; for details, see [1]).

**Figure 7 cancers-15-04286-f007:**
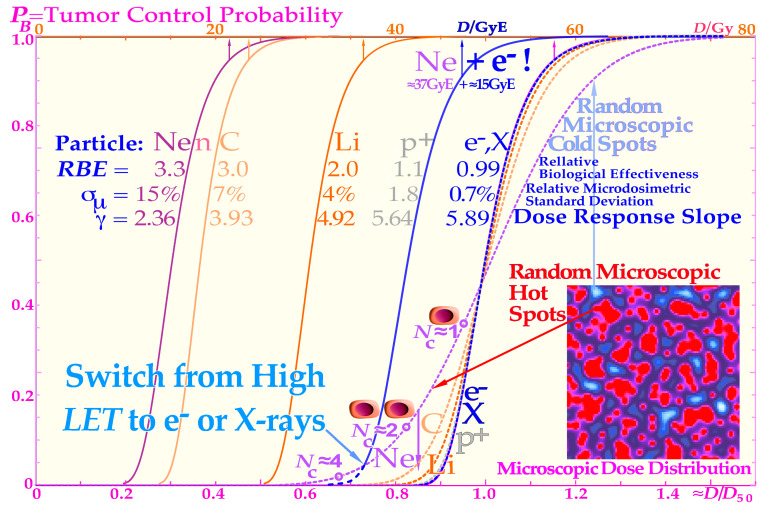
Description of the shape of the tumor control probability curve for a uniform cell line with different radiation modalities as a function of the absorbed dose (upper scale) and approximately normalized to the 50% tumor control dose (≈dose equivalent, lower scale, dashed lines) to more clearly see the effect on the *γ*_C_ value as the microdosimetric relative standard deviation increases with the LET. Not only are the hot spots often in the form of dual double-strand breaks (DDSBs, Figure 1; [1] (Figures 2 and 16)) and cold regions become more extreme with increasing LET, but also the RBE increases, thus reducing the total dose about threefold with carbon, neutron, and neon, increasing the relative standard deviation, and reducing the *γ*_C_ value more than desirable. For mixed high- and low-LET treatments such as neon ions + e^−^, a Gy-equivalent upper scale is needed in units: GyE.

**Figure 8 cancers-15-04286-f008:**
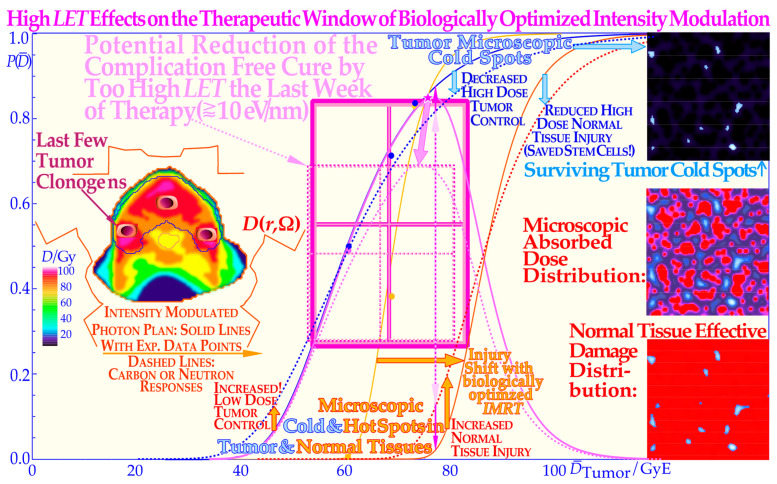
The adverse effect of a too-high LET on the complication-free cure reduces the tumor cure and simultaneously increases normal tissue injury (dashed lines, solid lines: [41,42] (Figure 4)). Interestingly, there is a cost-efficient clinical solution to this problem by switching to electrons, photons, or even protons during the last week of treatment. This will lead to a steeper tumor response [1] (Figures 20 and 22), [7,28], generating a higher complication-free cure, all at a lower delivered dose equivalent (see Figure 7 for Ne + e^−^) and reduced risk of damaging normal tissues, as demonstrated here. *P*($\bar{\;D}$) is the probability of tumor control or normal tissue damage as a function of the mean tumor dose.

**Figure 9 cancers-15-04286-f009:**
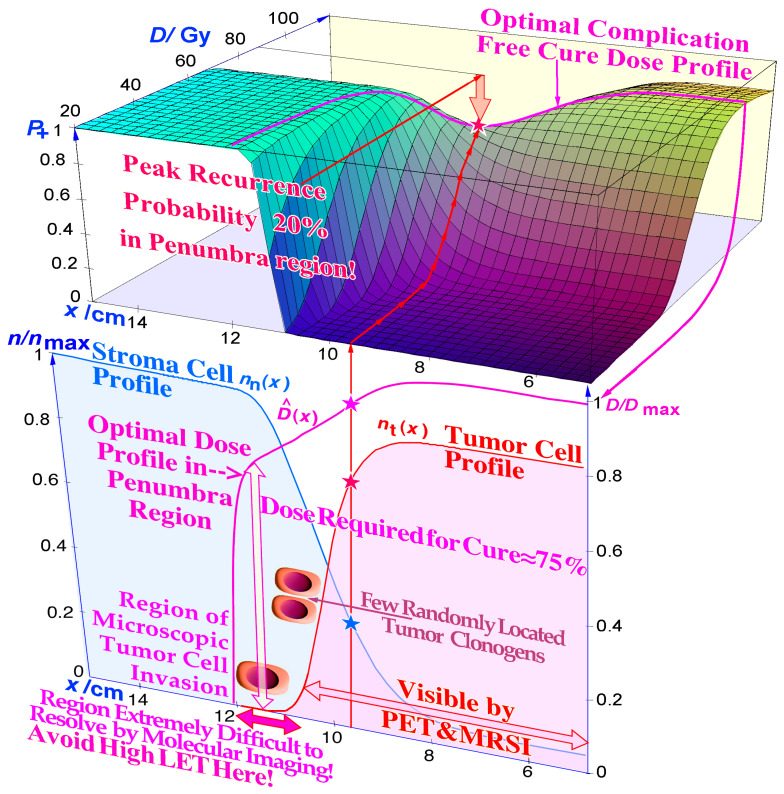
The optimal shape of the dose profile in the penumbra and setup margin region at the periphery of the internal target volume, assuming a microscopic invasive tumor of Gaussian spread (pink-shaded). Toward the end of the treatment, there are very few clonogenic tumor cells in this volume, and the probability that any of them is hypoxic is very small, even throughout the treatment, so there is no real need to use a high *LET* in this region (see also the periphery shown in Figure 10). It still needs a fairly high dose via low *LET* as the response is logarithmic (≈75–80% of *D*_max_) and much less if it initially received a high dose of ion therapy. *P*^+^ is the probability of a complication-free cure (Figure 8).

**Figure 10 cancers-15-04286-f010:**
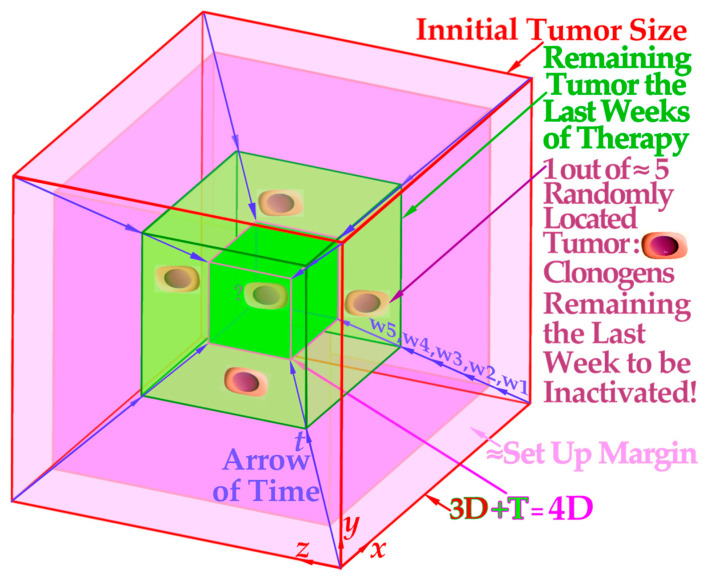
The projection of a 4D space–time internal target volume on a 2D flat surface shows the need for low-*LET* electrons or photons to round up an optimally performed light ion treatment. A 3D cube in 2D is two squares with all corners connected and a 4D cube in 3D is two 3D cubes with all their cubical corners connected, in this case with the blue fourth dimension time arrows. The periphery of the 4D internal target volume including the initial setup margin (pale pink; see Figure 9) and the few remaining gross tumor clonogenic cells (green volume) will substantially benefit from the last 10–15 GyE being delivered with minimal *LET* and microdosimetric variance of electron or photon beams. Interestingly, both the few remaining clonogenic tumor cells in the gross tumor and the setup margin are best eliminated with an optimized 15 GyE low-*LET* treatment round-up, and for bulky tumors, the first 5 GyE of those may include a concomitant higher-*LET* gross tumor boost [50].

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
