# Peer review of "TP53 and the Ultimate Biological Optimization Steps of Curative Radiation Oncology"

_cancers, 2023, doi:10.3390/cancers15174286_

Round 1

Reviewer 1 Report (Previous Reviewer 1)

The author has addressed the issue raised from the previous and the manuscript is ready for acceptance

Author Response

Thank you very much!

Anders

Reviewer 2 Report (Previous Reviewer 2)

In the revised manuscript, the author has addressed the comments and has followed some though not all suggestions. The additions and changes have improved and focused the manuscript although the structure still makes it difficult to read for the intended audience. Nevertheless, I consider it acceptable after some minor revisions.

Overview and specific comments to changes made in the revised manuscript:

The suggested references om PRIMA-1 and APR246 have been added to qualify the statements on their effect on p53.

The section on secondary cancer has been modified.

The author accepts the comment about the relevance of sections 2.5-2.7 and 3 and has removed the word “apoptosis” from the title but has decided against my other suggestion regarding publishing these sections separately.

Line 108-9: “Interestingly, the new DNA repair based formulation inherently describes LDHS and LDA as they are linked to the DNA repair system of most if not all normal tissues…”

Because side effects of radiotherapy occur in normal tissues, this implies to me that apoptosis is important for adverse normal tissue reactions. I think it should be pointed out that apoptosis is hardly causal for late normal-tissue damage.

Fig. 1 has been enlarged which improves the readability to some extent. It is a heavy start on the manuscript but it should probably be kept in the main text.

Fig. 2 has been moderately simplified (better now)

The old Fig. 3 has been deleted which helps focus the manuscript.

Fig. 3: RBE is of course dimensionless but the ticks on the right-hand side y-axis were not labelled with units. These have been added in the revised version.

Fig. 4 has no unit labels on the diagram axes. Presumably the graphs are based on curve fits so I would expect the ticks to be labelled. Or is it a qualitative, schematic diagram? The fractions f, g, h, I, n etc. with subscripts are not explained in the caption. What does “Totally ‘Mised’ Cells” mean? Should it be 'Missed'?

Fig. 8: P(D) on the y-axis is not explained in the caption: Presumably it covers Tumour Control Probability (TCP) and Normal-Tissue Complication Probability (NTCP)? Similarly, GyE is not defined anywhere (presumably Gray equivalent?). The reader should not have to guess this.

Furthermore, I have some general comments to the authors replies to my original comments:

I read the original manuscript according to title (TP53 apoptosis...) and the introduction. The latter was and still is very much based on the RHR model. The first parts of section 2 go through the history of inactivation models, highlighting the authors’ recent RHR model and drawing quite wide-ranging conclusions from it. My problem with that is that it is difficult to follow the derivations and the concepts without reading the two previous, long and complex papers in Radiation Research and so most readers will have to trust the authors’ statements with no real chance of assessing the conclusions. After going through the aspects cell survival, apoptosis, and the role of TP53, the rest of the manuscript appears disconnected because the clinical optimization and the ion beam microdosimetric considerations might also be published separately without reference to the RHR model. I would have expected these parts to be novel predictions from the model that might be tested. Instead they are more a discussion of the application of ion therapy with interpretations in terms of the RHR model.

Several mechanisms have been suggested that will lead to the L-Q model or close approximations but the fact that the L-Q model fits most cell survival curves in the dose range 2-8 Gy does not mean that these models are mechanistically correct. Similarly, the RHR model may fit survival and apoptosis data better than the L-Q model, especially in the low-dose region (0-1 Gy), and it may be a more useful model (the future will tell) but it still does not mean that the model is mechanistically correct. Although it may be justified to explore the predictions of the model in terms of a mechanistic interpretation, the findings should not be presented as certainties to readers who may not appreciate the finer details and the limitations of the model.

The statements below are, shall we say, at least very bold considering that they are based on interpretations of a model involving a number of assumptions and not on direct measurements:

Line 100: “This makes HR about 3 times more important than NHEJ at high LET’s…”

Line 104: “…allows quantification of concurrent independent NHEJ and HR repair as well as …”

Line 143: “…the recent DNA repair publication [2] demonstrated that this early cell loss before establishing full repair efficiency is mainly due to p53 LDA induction”

Line 313: “It is (still) striking how much data can be derived by the new (more) accurate cell survival and DNA repair formulation…”

In the reply to my comment on the second paper (RR 2022), you state that IHR (HR for ion-induced damage?) works in almost all phases (of the cell cycle). However, HR uses the sister chromatid strand as template for error-free repair. In the G1 and early S phase there are no (or only very few) sister chromatid strands so where do the templates come from in these cell-cycle phases?? In the same paper, the acknowledgement to M. Edgren in a footnote for providing survival data from a cell line (unnamed in the footnote) is of course fair to the person donating her data but a proper reference to the U1690 cell line and how the data were obtained would have been better.

Typo: In line 237, it should probably read HDA instead of HAD.

The English spelling needs editing, e.g. (not a complete list):

p. 2, lines 50-54 (and elsewhere): (Non-)’Homologous’ instead of ‘(Non-)Homologues’

p. 2, line 69: ‘breaks’ instead of ‘brakes’

p. 4, line 112: ‘whether’ instead of ‘weather’

p. 4, line 131: ‘pale’ instead of ‘pail’

p. 6, lie 168: ‘very’ instead of ‘vary’

etc.

Author Response

Because side effects of radiotherapy occur in normal tissues, this implies to me that apoptosis is important for adverse normal tissue reactions. I think it should be pointed out that apoptosis is hardly causal for late normal-tissue damage. Good! Added at line 131 Important Clinically, these ideas came out of your question!:

The LDA and LDHS of normal tissues are caused by about 5-15% acute low dose apoptosis (Figs 1 and 2 and [4, 5]), but interestingly, most likely due to the compensating measure of caspase 3 induced cellular repopulation [6] late effects are small. This will reestablish homeostasis in the normal tissues, and thus minimize late normal tissue damage, but it may sometimes also repopulate malignant tumor clonogens if they are not totally eradicated by the treatment [6]. This means that the LDA really protects normal tissues from potential low dose mutations before NHEJ and HR are fully functional and can take care of the damage (cf 2.2)! Furthermore, it means for radiation therapy that the fully functional DNA repair system should be continued to be utilized until the more severe high dose apoptosis sets in after about 2-2.5 GyE as seen in Fig 1. Already now we understand that there is an optimal radiation therapy fractionation window in normal tissues around 1.8-2.3 Gy/ Fr to minimize normal tissue damage as discussed in further detail below (cf 2.5).

Fig. 4 has no unit labels on the diagram axes. Added: fractional cell survival S(D) in text (like RBE dimension less). Presumably the graphs are based on curve fits so I would expect the ticks to be labelled. Or is it a qualitative, schematic diagram? The fractions f, g, h, I, n etc. with subscripts are not explained in the caption. They were in old fig 3 my mistake, added. What does “Totally ‘Mised’ Cells” mean? Should it be 'Missed'? yes  Un hit may be more clear! Changed in Fig! added:(plain f, g, k, l are repair fractions where HR fixes various NHEJ misrepair, fh plain HR and fn plain NHEJ cf [2, 14] for further details)

Fig. 8: P(D) on the y-axis is not explained in the caption: Presumably it covers Tumour Control Probability (TCP) and Normal-Tissue Complication Probability (NTCP)? Similarly, GyE is not defined anywhere (presumably Gray equivalent?). The reader should not have to guess this. Added: For a mixed high and low LET treatments such as Neon + e- a Gy-Equivalent upper scale is needed in units: GyE. In Fig 7 and: P(D) is the probability for tumor control or normal tissue damage as a function of the mean tumor dose. In Fig 8

I read the original manuscript according to title (TP53 apoptosis...) and the introduction. The latter was and still is very much based on the RHR model. The first parts of section 2 go through the history of inactivation models, highlighting the authors’ recent RHR model and drawing quite wide-ranging conclusions from it. My problem with that is that it is difficult to follow the derivations and the concepts without reading the two previous, long and complex papers in Radiation Research and so most readers will have to trust the authors’ statements with no real chance of assessing the conclusions. True! After going through the aspects cell survival, apoptosis, and the role of TP53, the rest of the manuscript appears disconnected because the clinical optimization and the ion beam microdosimetric considerations might also be published separately without reference to the RHR model. True! I would have expected these parts to be novel predictions from the model that might be tested. Instead they are more a discussion of the application of ion therapy with interpretations in terms of the RHR model.

Several mechanisms have been suggested that will lead to the L-Q model or close approximations but the fact that the L-Q model fits most cell survival curves in the dose range 2-8 Gy does not mean that these models are mechanistically correct. Similarly, the RHR model may fit survival and apoptosis data better than the L-Q model, especially in the low-dose region (0-1 Gy), and it may be a more useful model (the future will tell) but it still does not mean that the model is mechanistically correct. Although it may be justified to explore the predictions of the model in terms of a mechanistic interpretation, the findings should not be presented as certainties to readers who may not appreciate the finer details and the limitations of the model. True!

Line 100: “This makes HR about 3 times  (removed) more important than NHEJ at very (added) high LET’s…” ( This value was used by the algorithm to accurately describe measurements!)

Line 104: “…allows approximate (added) quantification of concurrent independent NHEJ and HR repair as well as …”   It is not surprising that NHEJ and HR repair are working at different locations inside the same cell nucleus and their actual values are  basically determined by their multivariate Poisson probability distribution (cf old fig 3!).

Line 143: “…the recent DNA repair publication (added:) [2: Figs 7, 9a and b, 12a, c, d] demonstrated that this early cell loss before establishing full repair efficiency is mainly due to p53 LDA induction” (added:) in general agreement with [4, 5] and the present Figs 1 and 6 for a TP53 intact tumor.                                                 This is why I initially got so focused on apoptosis! You can see in Fig6 that the shaded area for 40eV/nm is disappearing at low doses. Also for 60Co:  Apoptosis ≈ Survival as for 40eV/nm!!! ( violet and black curves almost coincides) But not for 80 & 160 eV/nm!

Line 313: “It is (still) striking how much data can be derived by the new (more) accurate cell survival and DNA repair formulation…” %). Changed to: It is striking how much data can be estimated by the new more flexible and thus probably more accurate cell survival and DNA repair formulation.

In the reply to my comment on the second paper (RR 2022), you state that IHR (HR for ion-induced damage? Inter Homolog Recombination spelled out in Fig 1, using the paternal allele for a maternal damage may sometimes work but generally more risky) works in almost all phases (of the cell cycle). However, HR uses the sister chromatid strand as template for error-free repair. In the G1 and early S phase there are no (or only very few) sister chromatid strands so where do the templates come from in these cell-cycle phases?? In the same paper, the acknowledgement to M. Edgren in a footnote for providing survival data from a cell line (unnamed in the footnote) is of course fair to the person donating her data but a proper reference to the U1690 cell line and how the data were obtained would have been better.

This manuscript is a resubmission of an earlier submission. The following is a list of the peer review reports and author responses from that submission.

Round 1

Reviewer 1 Report

The author has discussed recent developments in radiobiology and cancer treatment. However, the claims of the author is not complemented in the review of literature. There are only 3 publications cited from 2020-2022 and the rest are a couple of decades old. 

Moreover, such datasets should be targeted to clinical radiation journals rather than cancers

Author Response

The author has discussed recent developments in radiobiology and cancer treatment. However, the claims of the author is not complemented in the review of literature. There are only 3 publications cited from 2020-2022 and the rest are a couple of decades old. 

Unfortunately I am bringing forward new results in an area were very little was published lately, that is also why I am anxious to get it publish as it has important clinical consequences and will significantly improve the result of current radiation treatment methods. A hand full more recent refs were added!

Moreover, such datasets should be targeted to clinical radiation journals rather than cancers.

Sorry I got an invitation and the titel of your issue is perfectly fitting because my message is presenting a whole new way of thinking about cancer treatment principles by radiation therapy, and of course partly supported by new data! Furthermore, the new approaches and principles presented are applicable to a wide range of new approaches not just apoptosis but senescence and a number of repair gene knock outs and inhibitors and reactivating TP53 mutant tumors that are a severe clinical problem and simultaneously improving effective ROS  based treatments.

Reviewer 2 Report

The present manuscript is based on two recent papers by the same author. In the first paper (Radiat. Res. 194, 2020) the author developed a model for cell inactivation based on different types of DNA lesions: lesions that are ‘directly lethal’ (i.e. which cannot be repaired), and two types of repairable lesions termed ‘sublethal damage’ (not to be confused with the operational term ‘sublethal damage repair’ associated with split-dose recovery) that are processed by base damage repair and the two major DNA double-strand break (DSB) repair mechanisms, non-homology end joining (NHEJ) and homologous recombination (HR). It was suggested that non-homologically repairable lesions are relatively simple types of lesions induced by indirect radiation action on DNA (i.e. by reaction of radiation-induced aqueous free radicals formed near DNA) and repaired by one of several base damage repair mechanisms, including single-strand break repair by base excision repair, or by NHEJ whereas the HR-repairable lesions are more complex lesions, in particular dual DSBs or DSBs with additional damage (multiply damaged sites), that are formed by direct action (i.e. ionisation) in the DNA molecule itself. The former lesions would be more common for low-LET radiation whereas the latter would dominate for the dense ionization track of ion beams (especially in the Bragg peak). Basically, the “non-homologically Repairable-Homologically Repairable” (RHR) model expresses the surviving fraction, S, as the fraction of cells not receiving a directly lethal (irreparable) lesion plus the fraction of cells receiving ‘sublethal lesions’ that are correctly repaired. The repairable ‘sublethal lesions’ were considered to be processed by either by non-homology types of repair (base damage repair and NHEJ, i.e. non-HR) alone or HR repair alone, or by both simultaneously if occurring concurrently. The probability of survival was expressed as the sum of the Poisson probabilities for the individual components using cross sections for the induction and outcome of each type of lethal or repaired/unrepaired ‘sublethal lesion’ multiplied by the fluence and comprised five elements with 11 free parameters. The expression was converted in terms of dose and fitted to previously published experimental survival data for a lung epithelial cell line that displays low-dose hyper-radiosensitivity (LDHS), to mouse embryo fibroblasts with different knocked-out repair genes, to cell survival data for a Chinese hamster cell line in different cell-cycle phases, and to irradiation with ions of different LET.

In the second paper [Radiat Res 198, 2022], the model was expanded to include combinations of concurrent ‘sublethal lesions’ that were each repaired or misrepaired by non-HR mechanisms or HR alone, or non-HR mechanisms followed by HR. Because HR-repair is in principle error-free and can only occur after replication (i.e. in late S and G2 phase), non-HR repair cannot correct lesions misrepaired by HR and this combination was, therefore, not included. The modified approach resulted in eight different forms of misrepair and an expression with the sum of seven elements incorporating 15 free parameters, which by approximation could be reduced to five elements and 9 free parameters. The improved model was fitted to the same survival data for TP53-wildtype lung epithelial cells as above, to survival of the TP53-mutated SCLC cell line, U1690 cells, and to apoptotic fractions of TP53-wildtype AA melanoma cells. Of note, no information was given on the source of the U1690 data.

The present manuscript is a continuation of the two previous papers, applying the improved model to U1690 cells, in particular the LET dependency of apoptotic fractions and RBE for cell survival for these cells with and without the p53-‘reactivating’ drug PRIMA-1. Furthermore, considerations on the fractionation window based previously published data on a lung epithelial cell line, on secondary cancer induction (based on non-apoptotic misrepair), and clinical dose fractionation are included. The final section is a discussion on dose-response relationships based on microdosimetric characteristics of different ion beams.

General comments:

The manuscript covers a broad range of topics considering that the RHR model has not been validated and has only been applied to a narrow set of data so far. However, the high number of free parameters built into the model carries a considerable risk of overfitting the data. Although a good curve fit is a prerequisite for accepting a model it does not constitute proof of its mechanistic basis. A further problem is the lack of a clear distinction between assumptions and facts. A model may well be based on assumptions but these should be clearly stated and not presented as facts. The statements in the manuscript are not always supported by references and several statements are presented as facts although they are speculative. Frequently, the speculations are based on little or narrow experimental evidence and many findings are overinterpreted and generalized without justification. Most of the figures are overloaded and difficult to decipher for the reader. Frequently, symbols, labelling or units of axes, and basic captions are missing or ambiguous. The manuscript would benefit from making the figures simpler and clearer.

A comparison of a single TP53-mutated SCLC cell line with TP53 wildtype melanoma and lung epithelial cell lines to represent tumour and normal tissue, respectively, is a gross generalization of the effect of TP53-dependent apoptosis, especially since no apoptosis data are presented for the lung epithelial cell line. The claim that PRIMA-1 mediated upregulation of apoptosis is caused by reactivation of p53 may be questioned by the finding that PRIMA-1 increases apoptosis by upregulation of caspase-3 in TP53 wildtype and mutant cell lines (reviewed in [Perdrix et al., Cancers 9:172, 2017]) and that its active form PRIMA-1_met (APR246) has been reported to have p53-independent effects [Yoshikawa, Oncol Rep 35:2543-2552, 2016]. Furthermore, PRIMA-1 does not seem to reactivate p53's function as a transcription factor since it had little effect on transcription of CDKNA1 coding for the cyclin-dependent kinase inhibitor, p21, in esophageal squamous cell carcinoma cells [Furukawa, Cancer Sci 109: 412-21, 2018].

Regarding the risk of secondary cancer after radiotherapy, it is difficult to see how non-apoptotic misrepair in a melanoma cell line can inform on the risk of secondary cancer arising from co-irradiation of normal tissue. Although the RHR model predicts a maximum at 1-4 Gy of viable misrepaired cells consistent with conventional wisdom for secondary cancer induction, there is increasing evidence that solid secondary tumours do not show a maximum in this dose range but occur in regions that have received intermediate to high doses.

The Figures 3 and 5 have been published before [Radiat Res 198, 2022] but neither the reference nor permission to reproduce them are given. The sections 2.5-2.7 bear relatively little relation to the title, “TP53 apoptosis…” and several of the considerations are not really dependent on the RHR model (see also major specific comments). Section 3 is not necessary for the rest of the manuscript and the manuscript would be strengthened by publishing it separately.

The best evidence that the RHR may have some merit is probably Figure 5 of the previous paper [Radiat Res 194, 2022] showing the contributions of the different type of misrepair to inactivation of lung epithelial cells after low-LET irradiation. At low doses (0-2Gy), repair of single lesions by misrepaired non-HR (in particular NHEJ) followed by HR misrepair, misrepaired by NHEJ or HR only, and unrepaired non-HR lesions misrepaired by HR, dominate. At intermediate doses (2-5Gy), concurrent lesions misrepaired by non-HR or HR become more important, and at higher doses (>5 Gy) irreparable lesions, concurrent non-HR lesions misrepaired by HR, and concurrent non-HR lesions misrepaired by NHEJ followed by HR, gradually replace the previous types. This seems plausible according to the expected change in damage induction with increasing dose. However, the claimed accuracy of the order of 1% is hard to believe without data on parameter values for several different cell lines. Thus the claimed high accuracy of the RHR model is not substantiated and appears overstated. In summary, I do not find the manuscript in its present form acceptable for publication.

Major specific comments:

Abstract: “most TP53 intact normal tissues are LDHS and LDA”. This is a broad generalization based on a single lung epithelial cell line in which apoptosis was not even determined. Many normal cells undergo permanent cell-cycle arrest (senescence or differentiation) rather than apoptosis after irradiation. Furthermore, dose-limiting late reactions are not, or only very weakly, associated with clonogenic cell death. Instead, inflammatory reactions and interactions between different cell types, including immune cells, are important.

P. 2, line 67-8: It is not clear which protein is referred to: presumably the author means phosphorylation of TP53 at Ser-45? The statement that DYRK2 is involved is not documented in Fig. 9 of Ref. 1 nor in the reference [Nakamura et al., Cancer Sci 95:7-11, 2004]. Why focus on DYRK2 (apart from ATM), and what is the evidence?

Figure 1 is overloaded and partly unreadable.

P. 3-4, line 93-98: Phosphorylation of p53 protein may be fast but p53 levels are kept low by binding to MDM2 and proteasomal degradation. Irradiation causes dissociation from MDM2 leading to upregulation of protein levels but this takes longer (of the order of ~ 1h). NHEJ dominates for low-LET radiation but if p53 is recruited mainly by this type of lesion, how does p53 then stimulate repair of high-dose and high-LET local damage? 

Line 99-100: What is the evidence that “HR is about 3 times more important than non-HR at high LET”? Takahashi et al. [Radiat Res 182:338-44, 2014] found that NHEJ was more important than HR even for high LET.

Line 106-107: apoptosis does not determine late normal-tissue reaction.

Line 108-9: p53 certainly plays a role in repair but calling it the master of repair is exaggerated. p53 decides the fate of cells (transient arrest & repair; permanent arrest; apoptosis) whereas DSB repair choice is determined by the structure of lesion [Scully et al., Nat Rev Mol Cell Biol 20:698-714, 2019]. Furthermore, TP53-mutated cancer cells and even TP53-negative cells are capable of DSB repair.

Line 114: what is meant by “early”? – does it mean an early time point (which?) or perhaps a low dose?

Figure 2 is overloaded. It would be easier to read if the text elements were shown separate from the curves which could be marked with different signatures.

Figure 3 is basically a combination of Fig. 9 and Fig. 10 from the previous paper [Radiat Res 198, 2022]. Is the figure really necessary here? Anyway, the figures would be clearer if the mathematical expressions were shown separately and the experimental points and curves were explained by a figure legend. The source of the data is missing. Presumably, the reference is Meijer et al., Int J Radiat Biol 81:161-72, 2005 as given in the author’s previous paper? This needs to be included.

Fig. 3 and text on p7: The summation sign Sigma_A-N implies a summation over 14 elements but is performed for only nine elements (eight forms of misrepair plus direct inactivation). In any case, the orthography of the mathematical summation sign is formally incorrect. Furthermore, a detailed explanation of the curve fitting approach is missing.  Fig. 3 states that the RHR model was fitted to all experimental data for the apoptotic fraction 48h after irradiation of AA melanoma cells with a dose of 3Gy of different radiation qualities (four different LET values in the range 0.3-160 keV/µm). Exactly how was the fit performed - were all different variables shown in Meijer et al. [IJRB 81, 2005] included? Given that the kinetics of apoptosis  showed large differences for different LETs, why was 48h chosen as the time point? Any method for detecting apoptosis gives only a snapshot at a given time point but not the cumulative contribution of apoptosis to cell death. Thus dead cells with completely degraded DNA, and cells that have not yet entered apoptosis, will not be counted.

P. 7, last sentence at the bottom: the relation between apoptosis, RBE and hypoxic cell kill is unclear. How do they relate and is there independent evidence for the claims? Furthermore, it is stated that optimizing apoptotic death of tumour cells is desirable. However, this is not necessarily true as apoptotic cell death avoids inflammatory reactions. In fact, a major trend in current cancer therapy is to enhance anti-tumour immune reactions, and radiotherapy with large fraction sizes or intermediate to high single doses act releases cellular constituents that act as an adjuvant to stimulate such immunogenic cell death.

Fig. 4. The cell lines and the source of the experimental data are not given and the method of curve construction is unclear. The comparison of p53wt and p53mut cells uses different ion beams and, therefore, cannot be compared. ‘There are no units on the right-hand side RBE axis, and the inserted table as well as the abbreviations ‘KI’ and ‘NIRS’ are not explained.

P. 8: PRIMA-1 modifies SH-groups and prevents (mis)folding of p53. Although it restores apoptosis, this may be related to upregulation of the mitochondrial (intrinsic) pathway of apoptosis (see above). Modification of p53 by PRIMA-1 or its derivative may not influence cell-cycle control as transcriptional activation of p21 is not restored [Furukawa et al., Cancer Sci 109:412-21, 2018]. Is there evidence that it influences the role of p53 in repair? It should be noted that p53 has multiple functions and mutated p53 may show gain-of-function.

Fig. 5: is a reproduction of Fig. 6 in [Radiat Res 198, 2022] but no reference is given. It is not clear how these curves were derived and which experimental data were used.

P. 9-10. Results from a single lung epithelial cell lines cannot be generalized to cover all normal tissue reactions. The line of arguments in Section 2.5 does not rely on the RHR model but only on the LDHS phenomenon.

P. 10-11. Non-apoptotic misrepair is postulated to be related to secondary cancer but it is unclear if secondary cancer is assumed to originate from surviving clonogenic cells or cells that are permanently arrested. The implied maximum at 1-4 Gy is not in agreement with the increasing evidence that secondary solid tumours after radiotherapy arise in the intermediate- to high-dose volumes rather than in the low-dose regions as traditionally assumed.

P. 11-12: The clinical example does not rely on the RHR model and might as well be derived using the L-Q formalism.

Minor comments:

LET is a macroscopic quantity (mean value), so why is the unit eV/nm used instead of the conventional unit keV/µm? Using the unit related to nm dimensions implies energy deposition at this scale which is stochastic and should be characterized by a distribution.

Generally, figure captions should contain explanations of the axes and graphical elements. These are inadequate or missing from most figures. The figure numbers do not always correspond to the order they are mentioned in the text.

A number of minor typos and spelling mistakes need to be corrected.
